# Safety and effectiveness of 8 weeks of Glecaprevir/Pibrentasvir in challenging HCV patients: Italian data from the CREST study

Alessio Aghemo[1,2]*, Marcello Persico[3], Roberta D'Ambrosio[4], Massimo Andreoni[5], Erica Villa[6], Abhi Bhagat[7], Valentina Gallinaro[8], Giuliana Gualberti[8], Rocco Cosimo Damiano Merolla[8], Antonio Gasbarrini[9]

1 Department of Biomedical Sciences, Humanitas University, Pieve Emanuele, Italy, 2 Division of Internal Medicine and Hepatology, Humanitas Research Hospital IRCCS, Rozzano, Italy, 3 Dipartimento di Medicina Clinica Medica, Epatologica e Lungodegenza, AOU OO. RR. San Giovanni di Dio Ruggi e D'Aragona, Salerno, Italy, 4 Division of Gastroenterology and Hepatology, Foundation IRCCS Ca' Granda Ospedale Maggiore Policlinico, Milan, Italy, 5 University of Tor Vergata, Rome, Italy, 6 UC Gastroenterologia, Dipartimento di Specialità Mediche, Azienda Ospedaliera Universitaria di Modena, Modena, Italy, 7 AbbVie Inc., North Chicago, Illinois, United States of America, 8 AbbVie srl, Campoverde, Latina, Italy, 9 Medicina Interna e Gastroenterologia, Università Cattolica del Sacro Cuore, Fondazione Policlinico Universitario Gemelli IRCCS, Roma, Italy

* alessio.aghemo@hunimed.eu

**Data Availability Statement:** AbbVie is committed to responsible data sharing regarding the clinical trials we sponsor. This includes access to

## Abstract

### Introduction

Glecaprevir/pibrentasvir (G/P) has demonstrated high rates (>95%) of sustained virologic response at posttreatment Week 12 (SVR12) in treatment-naïve (TN) patients with hepatitis C virus (HCV) infection and compensated cirrhosis (CC). Here, in a key real-world subset of TN Italian patients with CC, we evaluated the effectiveness and safety of 8-week G/P treatment, including subgroups of interest such as those with genotype 3 (GT3) infection, elderly patients, and those with more advanced liver disease.

### Methods

Subanalysis of Italian patients enrolled in the CREST study. The full analysis set (FAS) included all patients enrolled in the study; the modified analysis set (MAS) excluded patients who discontinued G/P for nonvirologic failure or who had missing SVR12 results. Primary and secondary endpoints included SVR12 and safety, respectively.

### Results

Of 42 patients included in the FAS, 1 discontinued for unknown reasons, and 2 had missing SVR12 data, leaving 39 patients included in the MAS. At treatment initiation, 74% of patients had ≥1 comorbidity, and 62% were receiving concomitant medications, including some that may potentially interact with G/P. SVR12 was achieved in 100% of patients in the MAS, and in 95% in the FAS. In subgroups of interest, the proportion of patients achieving SVR12 in the MAS (and FAS) was: 100% (94%) for patients ≥65 years, 100% (86%) for GT3, and

anonymized, individual, and trial-level data (analysis data sets), as well as other information (eg, protocols, clinical study reports, or analysis plans), as long as the trials are not part of an ongoing or planned regulatory submission. This includes requests for clinical trial data for unlicensed products and indications. These clinical trial data can be requested by any qualified researchers who engage in rigorous, independent, scientific research, and will be provided following review and approval of a research proposal, Statistical Analysis Plan (SAP), and execution of a Data Sharing Agreement (DSA). Data requests can be submitted at any time after approval in the US and Europe and after acceptance of this manuscript for publication. The data will be accessible for 12 months, with possible extensions considered. For more information on the process or to submit a request, visit the following link: https://www.abbvie.com/our-science/clinical-trials/clinical-trials-data-and-information-sharing/data-and-information-sharing-with-qualified-researchers.html.

**Funding:** AbbVie sponsored the study, contributed to its design, and participated in the collection, analysis, and interpretation of the data and in the writing, reviewing, and approval of the abstract. All authors had access to all relevant data, and participated in the writing, review, and approval of the abstract. No honoraria or payments were made for authorship.

**Competing interests:** Aghemo A: Grant support from AbbVie and Gilead; Advisory board for Alfasigma, Gilead, Intercept, MSD, Mylan, and Sobi. Persico M: Consultant and speaker for AbbVie, Gilead, and MSD. D'Ambrosio R: Advisory board for AbbVie, Gilead, Takeda; Speaking and teaching: for AbbVie, Gilead, MSD; Research support from Gilead. Andreoni M: Board membership for AbbVie, Gilead, Merck, and ViiV; Grant from Merck; Speaker for BMS, Gilead, and Janssen. Villa E: Nothing to disclose. Bhagat A, Gallinaro V, Gualberti G, and Merolla RCD: Employees of AbbVie and may hold stock/options. Gasbarrini A: Consultant for AbbVie, Actial, Alfasigma, Eisai, Gilead, MSD, Sandoz, Sanofi, and Takeda. This does not alter our adherence to PLOS ONE policies on sharing data and materials.

100% (100%) for patients with platelet count <150 × $10^9$/L and FibroScan® >20 kPa. Overall, 2 (5%) patients had an adverse event and neither were serious.

## Conclusion

Results from this real-world Italian cohort demonstrated the safety and effectiveness of 8-week G/P, with SVR12 rate >95%, even in elderly patients. These findings further support real-world evidence of the use of short-course G/P treatment in all patients with CC, including those with GT3, and those with advanced liver disease.

## Introduction

Hepatitis C virus (HCV) is a major cause of chronic liver disease worldwide, and approximately 58 million people were estimated to live with HCV infection in 2019 [1, 2]. Highly effective and well-tolerated pangenotypic direct-acting antivirals (DAAs) have become the recommended treatment for most patients with chronic HCV infection [1]. The high cure rate of DAAs may allow the achievement of the World Health Organization's (WHO) goal of global HCV elimination by 2030 [2].

The prevalence of HCV in Italy has been estimated to be higher than anywhere else in Western Europe, particularly in elderly people, who are at a greater risk of developing chronic HCV infection, and are more likely to develop cirrhosis compared with younger patients [3–7]. A probabilistic model estimated that in January 2020 there were more than 400,000 individuals (prevalence of 0.68%) with chronic HCV infection in Italy, with approximately 73% asymptomatic and potentially undiagnosed or unlinked to care [4]. The remaining 27% of patients were estimated to have cirrhosis, which emphasizes the need to implement plans to improve screening and linkage to care of patients with HCV in Italy [4]. More recent screening studies conducted during the COVID-19 pandemic showed that depending on the population demographics the HCV prevalence varied from 0.07% up to 2.5%, and up to 29.4% of patients were not aware of their serological status [8–12].

Glecaprevir/pibrentasvir (G/P) is approved for the treatment of HCV in treatment-naïve (TN) patients with chronic HCV genotype (GT)1–6 infection without cirrhosis or with compensated cirrhosis (CC; Child-Pugh A) [13, 14]. Randomized controlled trials have shown that G/P therapy was well tolerated and led to high sustained virologic response at posttreatment Week 12 (SVR12) rates (≥93%) over treatment durations of 12 and 8 weeks [15–17]. In addition, real-world data, including Italian cohorts, have shown that G/P was effective (virologic cure ≥95%) and well tolerated even in patients traditionally classified as hard-to-treat [18–26].

The European Association for the Study of the Liver (EASL) highlighted that additional data are needed to further consolidate the effectiveness and safety of 8-week G/P treatment in TN patients with CC, with GT3 and/or with signs of portal hypertension (i.e., a liver stiffness >20 kPa with a platelet count <150 × $10^9$/L; according to the Baveno VI classification) [1].

The aim of this study was to evaluate the real-world safety and effectiveness of 8-week G/P therapy in Italian TN patients with CC, including subgroups of interest such as GT3, patients ≥65 years, and those with more advanced liver disease.

## Methods

### Study design and data collection

CREST is a retrospective noninterventional, multicenter observational study assessing data collected from TN patients with HCV and CC in Canada, Germany, Israel, Italy, France, and Spain, who initiated G/P on or after January 1, 2018. This subanalysis only included patients who received treatment in Italy; according with the local label, national or international recommendations, and/or local clinical practice; G/P treatment was decided before enrollment in the study. Data sources for eligible patients comprised all available medical records, including paper medical charts, electronic medical records, and clinical laboratory records. All data collected was anonymized so that the patient could not be identified. The study was carried out following the Good Clinical Practice guidelines, and according to the Declaration of Helsinki. The study was granted an exemption from obtaining informed patient consent and was approved by each of the institutional ethics committees for the sites in Italy.

### Patient selection

Eligible patients were enrolled in a chronological order by the local principal investigator. The inclusion and exclusion criteria for the CREST study have been previously published [27]. Briefly, patients were aged ≥12 years, diagnosed with HCV infection and with CC (documented as Child-Pugh A in the opinion of the investigator), TN to any approved or investigational anti-HCV DAA medication, and treated with G/P for 8 weeks. Patients were also excluded if they had any historical clinical evidence of hepatocellular carcinoma, decompensated cirrhosis, or events possibly related to liver decompensation including any current or past evidence of Child-Pugh B or C. The full analysis set (FAS) included all the patients enrolled in the study who were eligible as per inclusion and exclusion criteria, whereas the modified analysis set (MAS) excluded those patients who discontinued G/P for reasons other than virologic failure and/or who had missing data to document the primary endpoint.

### Endpoints

The primary endpoint was to evaluate the effectiveness of 8-week G/P therapy in TN patients with CC, as determined by SVR12, defined as HCV RNA less than lower limit of quantification 12 weeks post G/P treatment in the MAS. The secondary endpoint was to evaluate the safety and tolerability of 8-week G/P therapy by assessing the number of treatment emergent serious and nonserious adverse events (AEs), including changes in laboratory parameters of interest. Exploratory endpoints were to assess the effectiveness and safety of 8-week G/P in subgroups of interest such as patients with GT3, elderly patients, and those with more advanced liver disease (platelet count $<150 \times 10^9$/L and FibroScan® [EchoSens, Paris, France] >20 kPa). A window of 10–26 weeks posttreatment was deemed acceptable for the determination of the primary and secondary endpoints.

### Statistical analysis

Demographic and baseline characteristics were reported using descriptive statistics. Categorical variables were reported using frequency tables, and quantitative variables were reported as mean (± standard deviation) or median (interquartile range [IQR], Q1–Q3). The primary endpoint was assessed by the percentage and the exact Clopper-Pearson 95% [28] confidence interval of patients achieving SVR12; in the FAS population, this was calculated using nonresponder imputation. For quantitative secondary endpoints, the median (IQR, Q3–Q1) were computed.

## Results

### Patient disposition

In total, the CREST study enrolled 386 patients across 5 countries. This subanalysis included 42 patients enrolled in Italy, of which 1 (2.4%) discontinued for unknown reasons. Of the 41 patients who completed the study, 2 (4.8%) were unable to be assessed for SVR12 due to missing data (Fig 1). This resulted in 39 (93%) patients being included in the MAS.

### Demographics and baseline characteristics

Most patients included in the FAS were male (n = 31; 74%); median (range) age was 60 (36–85) years; 26 were aged <65 years (62%), 7 (17%) were aged ≥65 and <70 years, and 9 (21%) were aged ≥70 years (Table 1). HCV GT1 was the most common infection (n = 18; 43%), whereas GT3 was reported in 7 (17%) patients. Five (12%) patients initiated treatment without an established HCV GT. Nine out of 27 patients (33%) had FibroScan ≥20 KPa, and 16 out of 36 (44%) had platelets count $<150 \times 10^9$/L. Six out of 24 patients had platelets $<150 \times 10^9$/L and FibroScan >20 kPa.

At treatment initiation, 19 (45%) patients reported 1 comorbidity, and 12 (29%) more than 1. Although fewer patients aged ≥65 years reported having 1 comorbidity compared with

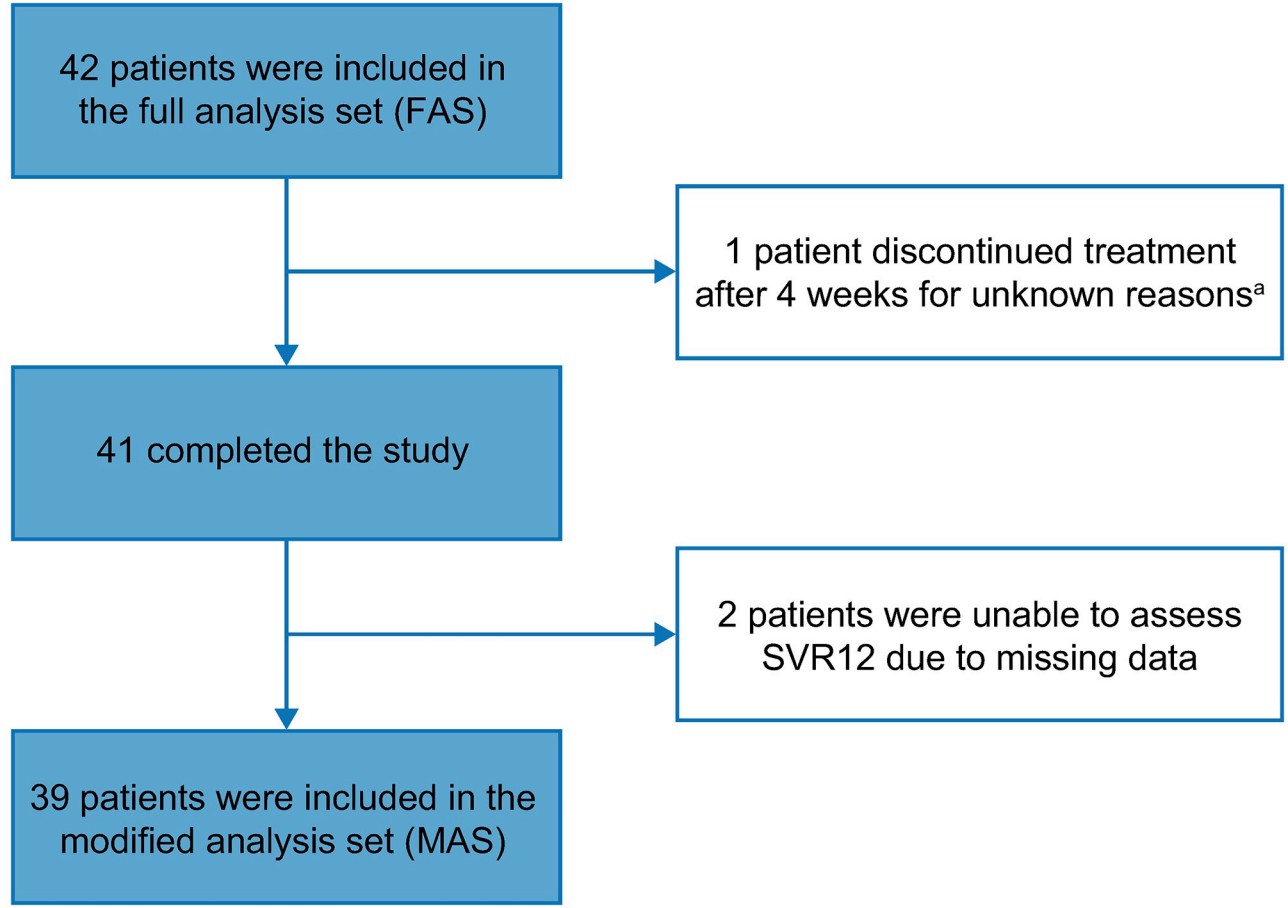

**Fig 1. Patient disposition.** [a]Although treatment was discontinued, this patient achieved SVR12. *SVR12* sustained virologic response at posttreatment Week 12.

**Table 1. Baseline demographics and clinical characteristics.**

| Characteristic | Patients N = 42 |
|---|---|
| Male | 31 (73.8) |
| Age, years | |
| median (range) | 59.5 (36–85) |
| <65 | 26 (61.9) |
| ≥65 and <70 | 7 (16.7) |
| ≥70 | 9 (21.4) |
| Race, white | 42 (100) |
| HCV genotype | |
| 1 | 18 (42.9) |
| 2 | 11 (26.2) |
| 3 | 7 (16.7) |
| <65 years | 6 (23.1) |
| ≥65 years | 1 (6.2) |
| 4 | 1 (2.4) |
| Unknown | 5 (11.9) |
| HCV RNA, median, (range), $\log_{10}$ IU/mL (n = 37) | 6.23 (2.78–7.99) |
| FibroScan® score, kPa, median (range) (n = 27) | 16 (8.6–27.7) |
| ≥20 (n = 27) | 9 (33.3) |
| <65 years (n = 16) | 5 (31.2) |
| ≥65 years (n = 11) | 4 (36.4) |
| Platelets, median (range), $10^9$/L (n = 36) | 158.5 (73.0–280.0) |
| Platelets <150 (n = 36) | 16 (44.4) |
| <65 years (n = 23) | 11 (47.8) |
| ≥65 years (n = 13) | 5 (38.5) |
| Platelets <100 (n = 36) | 6 (16.7) |
| Platelets <150 × $10^9$/L + FibroScan >20 kPa (n = 24) | 6 (25.0) |
| <65 years (n = 15) | 4 (26.7) |
| ≥65 years (n = 9) | 2 (22.2) |
| Bilirubin, median (range), mg/dL (n = 34) | 0.7 (0.3–1.9) |
| Albumin, median (range), g/dL (n = 29) | 4.1 (3.6–33.2) |
| ALT, median, (range), U/L (n = 36) | 96 (15–424) |
| Encephalopathy grade | |
| None | 40 (95.2) |
| Unknown | 2 (4.8) |
| Ascites | |
| None | 39 (92.9) |
| Slight | 1 (2.4) |
| Unknown | 2 (4.8) |

Data are n (%) unless otherwise specified.

FibroScan® is manufactured by EchoSens (Paris, France).

*ALT* alanine aminotransferase; *HCV* hepatitis C virus; *RNA* ribonucleic acid.

patients aged <65 years (38% vs 50%, respectively), the proportion reporting >1 comorbidity was higher in the elderly cohort (50% vs 15%). The two most common comorbidities were hypertension (23%; n = 6) and active drug use (12%, n = 3) in patients aged <65 years, and hypertension (56%, n = 9) and diabetes mellitus (31%, n = 5) in patients aged ≥65 years

**Table 2. Comorbidities and use of concomitant medications.**

| Characteristic | All patients | <65 years | ≥65 years |
|---|---|---|---|
| | N = 42 | n = 26 | n = 16 |
| **Comorbidities** | | | |
| 1 | 19 (45.2) | 13 (50.0) | 6 (37.5) |
| >1 | 12 (28.6) | 4 (15.4) | 8 (50.0) |
| None | 11 (26.2) | 9 (34.6) | 2 (12.5) |
| **Comorbidities present at treatment initiation**[a] | | | |
| Hypertension | 15 (35.7) | 6 (23.1%) | 9 (56.2%) |
| Diabetes mellitus | 6 (14.3) | 1 (3.8%) | 5 (31.2%) |
| Drug addiction | | | |
| Active user | 3 (7.1) | 3 (11.5%) | 0 (0.0%) |
| Former user | 2 (4.8) | 2 (7.7%) | 0 (0.0%) |
| Alcoholism | 2 (4.8) | 2 (7.7%) | 0 (0.0%) |
| Other liver disease | 2 (4.8) | 2 (7.7%) | 0 (0.0%) |
| Hyperlipidemia | 2 (4.8) | 1 (3.8%) | 1 (6.2%) |
| Hepatitis B | 2 (4.8) | 2 (7.7%) | 0 (0.0%) |
| Anxiety | 1 (2.4) | 0 (0.0%) | 1 (6.2%) |
| Psychiatric disorders | 1 (2.4) | 1 (3.8%) | 0 (0.0%) |
| Other | 13 (31.0) | 6 (23.1%) | 7 (43.8%) |
| **Use of concomitant medications** | | | |
| Yes | 26 (61.9) | - | - |
| No | 14 (33.3) | - | - |
| Unknown | 2 (4.8) | - | - |

Data are n (%)

[a]Multiple selection was allowed in the case report form for this variable, so aggregate percentage may exceed 100%.

(Table 2). Overall, 26 (62%) patients were receiving concomitant medications at treatment initiation. The most frequent medication was acetylsalicylic acid (n = 5; 19%), followed by bisoprolol and metformin (n = 4; 15% each; S1 Table). Out of 45 different medications coadministered with G/P, 28 were predicted to have no interaction, 6 were deemed to have potential interaction, 9 to have potential weak interaction, and 2 (atorvastatin and simvastatin) were suggested to not be coadministered with G/P, according to the University of Liverpool HEP Drug Interactions Checker [29].

## Efficacy

All patients in the MAS achieved SVR12 (n = 39/39), whereas in the FAS SVR12 was achieved by 95% (n = 40/42) of patients (Fig 2). One patient in the FAS who achieved SVR12 discontinued the treatment after 4 weeks. Of the 2 patients in the FAS who had missing SVR12, one had GT2 and one GT3. The percentages of patients achieving SVR in the MAS (and FAS), stratified by subgroups of interest, were: 100% (94%) for patients aged ≥65 years, and 100% (96%) for patients aged <65 years. Irrespective of the age group, all patients with platelet count <150 × $10^9$/L and FibroScan >20 kPa, achieved SVR12, both in the MAS and in the FAS.

## Safety

Of 42 patients in the FAS, 2 (4.8%) patients experienced an AE (fatigue and asthenia [n = 1; 2.4% each]) neither of which were study-drug–related or lead to study-drug discontinuation

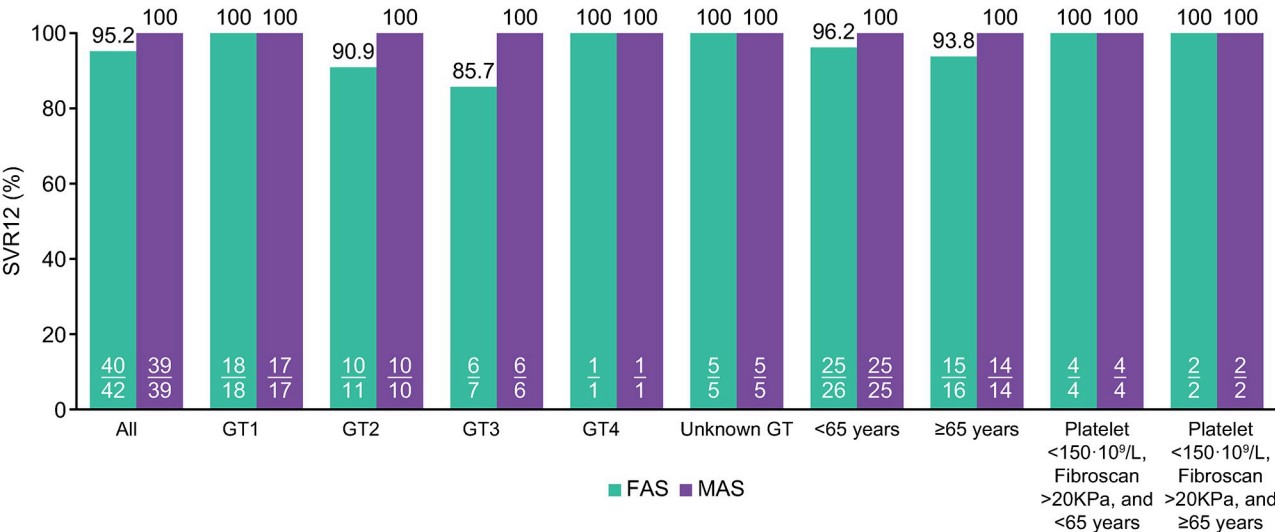

**Fig 2. SVR12 by genotype and subgroups of interest.** *FAS* full analysis set; *GT* genotype; *MAS* modified analysis set; *SVR12* sustained virological response 12 weeks post-treatment.

(Table 3). No patients reported having a serious AE, and no laboratory abnormalities were reported.

## Discussion

In the latest guidelines for treatment of HCV, EASL highlighted that additional results are needed to consolidate the recommendations for 8-week G/P in GT3 TN patients with CC, and in patients with CC and signs of portal hypertension (i.e., a liver stiffness >20 kPa with a platelet count $<150 \times 10^9$/L) [1]. This suggests that real-world data on the use of 8-week G/P in patients with HCV who are considered difficult-to-treat are limited. Here, we provide some data to alleviate this gap.

This retrospective real-world study reported the effectiveness and safety of 8-week G/P in 42 patients with chronic HCV infection treated in Italy. All patients included in the MAS

**Table 3. Adverse events.**

| Patients with AEs | Patients (N = 42) |
|---|---|
| Any AE | 2 (4.8) |
| Serious AEs | 0 (0) |
| AEs leading to discontinuation of study drug | 0 (0) |
| Any drug-related serious AEs | 0 (0) |
| Common AEs (occurring in ≥1% and <10% of patients) | |
| Fatigue | 1 (2.4) |
| Asthenia | 1 (2.4) |
| Other | 1 (2.4) |
| ALT > 5 x ULN | 0 (0) |
| AST > 5 x ULN | 0 (0) |

Data are n (%).

*AE* adverse event; *ALT* alanine aminotransferase; *AST* aspartate aminotransferase; *ULN* upper limit of normal.

population with GT1–6 infection achieved SVR12, demonstrating high reproducibility of clinical trials' and real-world studies' results which have shown that both 8-week and 12-week G/P yields SVR12 rates ≥99 in the per-protocol population [16, 20, 24, 26]. Shortened treatment durations from 12 to 8 weeks can reduce healthcare costs, simplify the treatment pathway, and potentially enable the treatment of more patients over time [30].

EASL guidelines recommend that pangenotypic HCV drug regimens, including G/P, can be used to treat individuals without identifying their HCV genotype and subtype, in order to simplify therapy on a global scale [1]. Notably, 12% of the patients initiated treatment without an established HCV GT; all of these patients achieved SVR12. Furthermore, most patients in this study with known genotype had HCV GT1 or GT3, which is in-line with the worldwide prevalence, suggesting this was a representative population [31].

All patients in subgroups of interest such as patients aged ≥65 years and those with more advanced liver disease achieved SVR12 (MAS), supporting the use of 8-week therapy in these patient populations. This is in-line with previous findings showing that patients with evidence of portal hypertension achieved SVR12 ≥95% [32]. Similar to a previous Italian real-world study, age does not seem to be a predictor of treatment failure [24]. A post-hoc analysis, which includes 9 clinical trials, also showed that >95% of elderly patients receiving 8-week G/P achieved SVR12 [33].

The favorable safety profile observed in the current study is consistent with previous findings from clinical trials and real-world studies of G/P in patients with CC [15, 16, 21]. Two patients experienced an AE, neither of which were related to the study drug or led to study-drug discontinuation. None of the patients experienced serious AEs, and no laboratory abnormalities were reported. Such an encouraging safety profile, even in patients with CC, has the potential to allow patients to complete the treatment and therefore to increase the likelihood of cure.

Consistent with other studies reporting a considerable proportion of patients with comorbidities [20, 24], 73.8% of the patients reported at least one comorbidity, and those aged ≥65 years had at least two comorbidities more frequently than younger patients (50% vs 15%, respectively). The proportion of patients with current or former drug abuse was lower than expected (12%) compared with epidemiological data from Italy which estimated that 49% of patients with HCV are people who inject/injected drugs [4]. The proportion of patients with alcohol dependence was also lower (4.8%) than previously reported [23, 34].

Overall, 62% of the patients were receiving concomitant medications at treatment initiation. The majority of the coadministered medications were predicted to have no interaction with G/P, while 9 were expected to have a potential interaction and 2 (atorvastatin and simvastatin) were suggested not to be coadministered with G/P [29]. These results suggest that in real-world clinical practices, the coadministration of G/P with drugs with potential interactions has no obvious impact on safety. Although it is recommended to conduct a thorough drug–drug interaction assessment prior to a patient initiating treatment [1], interactions between DAAs and most nonprescribed drugs is limited [35], with potential drug–drug interactions often based on predictions or modeling of interaction pathways [29].

There are a few limitations to this study. Given the nature of retrospective chart reviews, not all information was readily available for all patients; therefore, in some instances individuals lost to follow-up lacked a documented reason, making it challenging to entirely evaluate the barriers to treatment completion. Patients with diagnosed decompensated Child-Pugh B or C cirrhosis were excluded from this study because treatment with G/P is not recommended or contraindicated. The cirrhosis status of patients was reported by physicians, although the methodology to assess cirrhosis was documented for every patient from the medical charts.

## Conclusions

The prevalence of HCV in Italy is the highest in Western Europe, with a high proportion of elderly patients. Advanced age can make treatment for HCV challenging due to multiple comorbidities and higher pill burden, making this patient population potentially challenging to treat. However, this real-world analysis of an Italian cohort demonstrates that short-course G/P treatment of 8 weeks was well tolerated and effective in TN patients with HCV infection and CC, including elderly patients. The use of DAAs makes it possible to achieve excellent results even in those patients who were traditionally considered difficult to treat. Indeed, these findings further extend real-world evidence, and confirm preliminary results on the effectiveness and safety of 8-week G/P treatment in all patients with CC, including special subgroups of interests such as those with GT3, and those with portal hypertension.

## Supporting information

**S1 Table. Concomitant medications and their expected interaction with glecaprevir/pibrentasvir.**
(DOCX)

## Acknowledgments

### Authorship

All named authors meet the International Committee of Medical Journal Editors criteria for authorship for this article, take responsibility for the integrity of the work as a whole, and have given their approval for this version to be published.

### Medical writing, editorial, and other assistance

The authors would like to express their gratitude to the patients who participated in this study and their families, as well as the study investigators and coordinators of the study. Glecaprevir was identified by AbbVie and Enanta. Medical writing support was provided by Marta Rossi, PhD, and Sean Littlewood, MRes of Fishawack Communications, Ltd, and funded by AbbVie.

## Author Contributions

**Conceptualization:** Abhi Bhagat, Valentina Gallinaro, Giuliana Gualberti, Rocco Cosimo Damiano Merolla.

**Investigation:** Alessio Aghemo, Marcello Persico, Roberta D'Ambrosio, Massimo Andreoni, Erica Villa, Antonio Gasbarrini.

**Supervision:** Abhi Bhagat, Valentina Gallinaro, Giuliana Gualberti, Rocco Cosimo Damiano Merolla.

**Writing – review & editing:** Alessio Aghemo, Marcello Persico, Roberta D'Ambrosio, Massimo Andreoni, Erica Villa, Abhi Bhagat, Valentina Gallinaro, Giuliana Gualberti, Rocco Cosimo Damiano Merolla, Antonio Gasbarrini.

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
