## [Decision Letter · Decision Letter 0]

2 Nov 2022

PONE-D-22-27602Safety and effectiveness of 8 weeks of Glecaprevir/Pibrentasvir in challenging HCV patients: Italian data from the CREST StudyPLOS ONE

Dear Dr. Aghemo,

Thank you for submitting your manuscript to PLOS ONE. After careful consideration, we feel that it has merit but does not fully meet PLOS ONE’s publication criteria as it currently stands. Therefore, we invite you to submit a revised version of the manuscript that addresses the points raised during the review process.

We look forward to receiving your revised manuscript.

Kind regards,

Umberto Vespasiani-Gentilucci, PhD.

Academic Editor

PLOS ONE

https://pubmed.ncbi.nlm.nih.gov/35543964/

In your revision ensure you cite all your sources (including your own works), and quote or rephrase any duplicated text outside the methods section. Further consideration is dependent on these concerns being addressed.

“Aghemo A: Grant support by AbbVie and Gilead; Advisory board for Alfasigma, Gilead, Intercept, MSD, Mylan, and Sobi.

Persico M: Consultant and speaker for AbbVie, Gilead, and MSD.

D’Ambrosio R: Advisory Board: AbbVie, Gilead, Takeda; Speaking and teaching: AbbVie, Gilead, MSD; Research support: Gilead

Andreoni M: Board membership for AbbVie, Gilead, Merck, and ViiV; grant from Merck; and speaker for BMS, Gilead, and Janssen.

Villa E: Nothing to disclose.

Bhagat A, Gallinaro V, Gualberti G, and Merolla RCD: Employees of AbbVie and may hold stock/options.

Gasbarrini A: Consultant for AbbVie, Actial, Alfasigma, Eisai, Gilead, MSD, Sandoz, Sanofi, and Takeda.”

Reviewers' comments:

Reviewer's Responses to Questions

**Comments to the Author**

1. Is the manuscript technically sound, and do the data support the conclusions?

Reviewer #1: Yes

Reviewer #2: Yes

2. Has the statistical analysis been performed appropriately and rigorously? 

Reviewer #1: Yes

Reviewer #2: Yes

3. Have the authors made all data underlying the findings in their manuscript fully available?

Reviewer #1: Yes

Reviewer #2: Yes

4. Is the manuscript presented in an intelligible fashion and written in standard English?

Reviewer #1: Yes

Reviewer #2: Yes

5. Review Comments to the Author

Reviewer #1: Thank you for the opportunity to review this article.

In this work, the authors evaluated the real-world safety and effectiveness of 8-week glecaprevir/pibrentasvir therapy in Italian treatment-naïve patients with compensated cirrhosis, analysing data in certain difficult to treat subgroups of interest.

The paper is well written and adds new important data to the real-world experience with the drug, specifically in the Italian context.

Minor comments:

1. I think that study population is not big enough to sustain a robust message on the efficacy and safety of the drug primarily in GT3 patients (7/42) and in patients with signs of portal hypertension. I think that key messages of conclusions (abstract and text) should be re-modulate shedding light mainly on the core message of the work (real-world safety and effectiveness of short course treatment in cirrhotic patients) and on the need to extend the real-world cohort (Italian and beyond) to confirm the preliminary data that appear to confirm safety and effectiveness of the drug also in special subgroups (GT3, portal hypertension).

2. Moreover, Table 3 does not add as much to the work. At best, it could be considered as supplementary material.

Best regards,

Paolo Gallo, M.D.

Clinical Medicine and Hepatology Unit

Department of Medicine

Fondazione Policlinico Campus Biomedico, Rome

Reviewer #2: Dear Editor,

Aghemo et al wrote a paper on a sub-analysis of a large retrospective observational non interventional multicentric study conducted in six countries worldwide to evaluate in a real-life setting the efficacy and safety of the Glecaprevir/Pibrentasvir direct antiviral agents (DAA) combination for the treatment of HCV-related compensated cirrhosis (CC). The present paper is focused on the Italian sub-cohort of patients included. Even if, nowadays, there are several reports on the efficacy of such drugs in real life settings, some concerns (mostly preconcepts) remain on the safety of NS3-containing DAA combinations, particularly in regards of safety and DDI (drug-drug interactions). For these reasons, every report on the real-life performance of such drugs is welcome and has a potential high interest to the readership of your journal. The paper reports high efficacy and safety both on per-protocol and intention-to-treat analysis in the 42 patients with CC included.

Moreover, the paper is lean and well-written in English.

There are only few minor issues to discuss:

Minor Issues:

1) Introduction section, line 69 and beyond: “A probabilistic model estimated that in January 2020 there were more than 400,000 individuals with chronic HCV infection in Italy, with approximately 73% asymptomatic and potentially undiagnosed or unlinked to care”.

There are several reports published in Liver International journal earlier this year that somewhat seem disavowing this projection, please briefly comment this fact and cite these papers (10.1111/liv.15314 ; 10.1111/liv.15273 ; 10.1111/liv.15316; 10.1111/liv.15375).

2) Conclusions section, line 242 and beyond: it could be useful to the aims of the paper to insert a brief sentence in the conclusions on the fact that, with these types of treatment, it could have to be pointed out that the so-called "difficult-to-treat" patients are definitely no longer to be considered in this way .

Kindest regards

6. PLOS authors have the option to publish the peer review history of their article (what does this mean?). If published, this will include your full peer review and any attached files.

Reviewer #1: No

Reviewer #2: No

---

## [Author Response · Author response to Decision Letter 0]

7 Dec 2022

Please see response to reviewers uploaded as a word document: "Response to Reviewers"

---

## [Editor Report · Decision Letter 1]

21 Dec 2022

Safety and effectiveness of 8 weeks of Glecaprevir/Pibrentasvir in challenging HCV patients: Italian data from the CREST Study

PONE-D-22-27602R1

Dear Dr. Aghemo,

We’re pleased to inform you that your manuscript has been judged scientifically suitable for publication and will be formally accepted for publication once it meets all outstanding technical requirements.

Kind regards,

Umberto Vespasiani-Gentilucci, PhD.

Academic Editor

PLOS ONE

Additional Editor Comments (optional):

None